# Quantification of Internal and External Load in School Football According to Gender and Teaching Methodology

**DOI:** 10.3390/ijerph17010344

**Published:** 2020-01-03

**Authors:** Juan M. García-Ceberino, Antonio Antúnez, Sebastián Feu, Sergio J. Ibáñez

**Affiliations:** 1Optimization of Training and Sports Performance Research Group (GOERD, acronym in Spanish), University of Extremadura, 10003 Cáceres, Spain; antunez@unex.es (A.A.); sfeu@unex.es (S.F.); sibanez@unex.es (S.J.I.); 2Faculty of Education, University of Extremadura, 06006 Badajoz, Spain; 3Faculty of Sports Science, University of Extremadura, 10003 Cáceres, Spain

**Keywords:** physical–physiological demand, Tactical Games Approach, Direct Instruction, physical education, inertial device

## Abstract

The design of teaching tasks determines the physical and physiological demands that students are exposed to in physical education classes. The purpose of this study is to quantify and compare, according to gender and teaching methodology, the external (eTL) and internal (iTL) load resulting from the application of two programs that follow different teaching methodologies, i.e., a Tactical Games Approach (TGA) and Direct Instruction (DI), to teach school football. The Ratings of Perceived Exertion (RPEs) recorded in the assessments were also studied. A total of 41 students in the fifth year of primary education from a state school from Spain participated in the study (23 boys and 18 girls), aged from 10 to 11 (*M* ± *SD*, 10.63 ± 0.49 years) and divided into two class groups. All the sessions were monitored with inertial devices that made it possible to record physical activity and convert the information into kinematic parameters. The results indicated that the students who followed the TGA method recorded higher iTL values (heart rate) and spent more time performing high-intensity activities. Boys recorded higher eTL, iTL, and RPE values than girls. There was an evolution in the RPE between the assessments, with both groups presenting a more efficient RPE in the posttest. The TGA method favors student physical fitness and health, thus, this method is recommended when planning physical education sessions.

## 1. Introduction

Teaching methodology has become the cornerstone upon which sports teaching is based [1]; thus, physical education is taught on the basis of different teaching methodologies [2]. It is the requirement of the physical education teacher to select the methodology that makes it possible for students to acquire the desired skills [3].

The scientific literature presents two main approaches to the teaching of contact sports (i.e., sports where a team tries to invade the opponent’s space in order to introduce the ball into the goal, basket, etc.): Teacher-Centered Approaches (TCAs) or techniques, and Student-Centered Approaches (SCAs) or tactics [4]. TCAs are characterized by the predominant role played by the teacher in the teaching–learning process. Among TCAs, the Direct Instruction (DI) method is the most common [5]. The DI method is based on the principle of technical competence. Initially, tasks are set that are isolated and decontextualized from the game, to be later incorporated into it (tactical problems) [6]. The teacher provides the initial information, using descriptive feedback to favor the successful performance of the tasks, while the students listen and execute the tasks according to the determined guidelines [7]. Specific exercises and unspecific simple games are the teaching means most commonly used in the DI method [8]. Teaching contact sports using TCAs presents some disadvantages, like a lack of motivation or the little time devoted to enhancing decision-making skills [9]. For this reason, new approaches have arisen, like the SCAs where the student becomes the center of the teaching–learning process. The Tactical Games Approach (TGA) stands out among SCAs, taking as its starting point the proposal for Teaching Games for Understanding [10]. The TGA method is based on game situations that are as contextualized and motivating as possible, or are in situated learning [11]. The tasks are presented in the form of small-sided games (SSGs) that represent tactical problems that have to be resolved by the students using their prior experience and their reflections on practice, thus encouraging decision-making [12]. The teacher uses interrogative feedback so that students develop decision-making skills and create their own tactical awareness [7].

The methodology used by the teacher in the sessions conditions the physical fitness of the students, as they carry out most of their physical activity in school [13]. Currently, despite the advent of SCAs, traditional direct instruction methods like TCAs are still the most frequently used in physical education, leading to high levels of inactivity among students (e.g., short time of motor practice) [14]. According to Roberts et al. [15], high levels of physical inactivity are the result of several factors: (i) the large amount of time taken up by teaching management; (ii) too much time being centered on the practice of technical skills; and (iii) approaches using complete versions of the game (in football play situations of 8 vs. 8, 11 vs. 11, etc.). Physical activity levels may be higher in contact sports like football, basketball, hockey, etc. [16]. The use of SCAs by teachers can help students to attain adequate levels of physical activity in their physical education classes due to the time spent on the game [17]. Therefore, these methods enhance the physical fitness of students, since they improve aerobic fitness [18]. In this regard, improved aerobic fitness is related to health enhancements [19].

In the practice of contact sports, girls show lower levels of physical activity than boys [20]. This could be due to the fact that boy are more traditionally linked to competition sports than girls [21]. Increasing the perceived competence and physical activity of girls is a challenge for physical education teachers [16].

Research that has analyzed the differences among the aforementioned methodologies has focused on variables related to the understanding of the game, such as declarative knowledge or decision-making. It has also studied psychological variables like motivation; the TGA method presents improvements in all these variables in comparison with the DI method [22,23]. However, few studies have analyzed the physical–physiological demands placed upon students in primary education after the implementation of different methodologies. Physical–physiological demands can be measured using external (eTL) and internal (iTL) workloads [24]. eTL refers to the physical demands placed upon the students, that is, those that can be observed from their behavior, while iTL refers to the physiological demands [25]. Likewise, Buchheit et al. [26] define eTL as the mechanical and locomotor stress caused by an activity, classifying it into kinematic and neuromuscular loads. Kinematic load analyzes displacements and their intensity, both outdoors [27] and indoors [28]. In contrast, neuromuscular load analyzes the forces exerted by the player as a result of the interaction with gravity and partners/opponents [29]. Fox et al. [30] define iTL as the physiological reaction and stress experienced by a stimulus which can be measured on a physiological (e.g., heart rate (HR)) or a psychological level (perception of effort, etc.) [31]. A detailed analysis of eTL and iTL provides a better understanding of the physical and physiological demands that students are subjected to during football practice [32].

The scientific literature presents different instruments for quantifying eTL and iTL, both objectively and subjectively. For example, inertial movement devices can be used for the objective quantification of eTL [33], as they combine a multitude of sensors (accelerometers, gyroscope, magnetometer, GPS, etc.), which facilitate the analysis and monitoring of loads [34]. The use of inertial movement devices makes it possible to analyze, in a valid and reliable manner, the kinematic (distances, velocities, sprints, etc.) and tactical [31,32,35], as well as neuromuscular (player load, impacts, etc.) [36,37] effort required to play football. Similarly, to quantify eTL subjectively, one can use the Integral System for the Analysis of Training Tasks (SIATE in its Spanish acronym); this system studies the pedagogical and organizational variables that define a task, and also makes it possible for teachers to quantify the subjective eTL of the tasks used to teach contact sports (eTL task = sum of the assigned value, from 1 to 5 points, of each of the six categories of ordinal load variables: degree of opposition, density of the task, percentage of simultaneous performers, competitive load, game space, and cognitive implication) [38].

The variable commonly used to study objective iTL is HR, which indicates the intensity of the exercise [39]. HR is obtained using inertial devices and heart rate monitors. Similarly, the Rating of Perceived Exertion scale (RPE) is also used to study iTL subjectively [40]. The students can use the RPE scale to indicate how tired they are during sports activities. In contact sports, a strong correlation has been observed between loads measured with HR and the RPE [41,42].

Figure 1 shows a diagram of the classification of workloads and the measuring instruments used.

Reina et al. [43] confirmed the existence of a relationship among the quantification of the subjective eTL and the eTL (Player Load) and the iTL (HR). Gómez-Carmona et al. [44] also confirmed that subjective eTL influences objective eTL, and found a high correlation between them.

The majority of studies on physical–physiological demands (eTL and iTL) were developed in the context of sports training to analyze official competitive matches [32,45] or training sessions using SSGs [25,46]. There are also studies that compared physical–physiological demands between SSGs and official competitive matches [35,47]. Few studies have comapred eTL and iTL in the context of a school. González-Espinosa et al. [13] compared the eTL and iTL resulting from the application of two programs using different methods, TGA and DI, in teaching basketball. These authors identified the TGA method as having better results in terms of the eTL and iTL variables, favoring a greater development of physical fitness among students. Furthermore, the students who participated in the program that followed the TGA method recorded better performance indicators in play.

Research centered on comparing teaching methodologies shows the need to complement studies analyzing variables, such as declarative and procedural knowledge, psychological parameters, and performance in play, with others that measure the physical–physiological demands placed on students. Thus, this study sought to quantify and compare, according to gender and teaching methodology, the eTL, iTL, and RPE resulting from the implementation of two intervention programs for teaching school football. Each intervention program was based on a different teaching methodology, i.e., TGA or DI. Our study hypotheses were the following: (h1) the TGA method will cause higher levels of eTL, iTL, and RPE than the DI method; and (h2) boys will record higher levels of eTL, iTL, and RPE than girls.

## 2. Materials and Methods 

### 2.1. Design

The present study was an empirical investigation with a manipulative strategy of a quasi-experimental, pre-experimental and longitudinal type, using a group pretest and posttest [48]. The physical–physiological demands recorded after implementing two intervention programs on school football teaching based on two different methodologies were analyzed. Both intervention programs comprised 11 sessions, including the assessment tests (pretest and posttest).

### 2.2. Sample and Setting

*Students*. A total of 41 students (23 boys and 18 girls), aged between 10 and 11 years (*M* ± *SD*, 10.63 ± 0.49 years) and distributed in two class groups from the 5th year of primary education in a state school in the central-west region of Spain, participated in the study. The distribution of the students by gender in each class in the Spanish state system is mixed and heterogeneous, and is organized by the school’s academic authorities. The education system prohibits segregating students by gender. Each group participated in a different intervention program: fifth year group A (8 boys and 12 girls) in the Tactical Games Approach Soccer (TGAS) program, centered on the learning of tactics; and fifth year group B (15 boys and 6 girls) in the Direct Instruction Soccer (DIS) program, centered on the learning of technique. The implementation of the programs in the class groups was random. No student participated in both intervention programs. The students had no previous contact with the contact sport of football in their physical education classes, although 15% of the students who participated in the TGAS program (group A) and 42.9% of the students who participated in the DIS program (group B) practiced football as an out-of-school activity during the application of the programs. All students who practiced football in an out-of-school context were boys. The groups were not modified to maintain the ecological validity of the study.

This study was included in the school curricular program after prior approval by the school council. It was carried out in accordance with the ethical guidelines of the 1975 Declaration of Helsinki and Organic Law 15/1999 of 13 December on the protection of personal data (LOPD) (BOE, 298, 14 December 1999) in order to guarantee the ethical considerations of scientific research with human subjects. Approval was also requested from the University Bioethics Committee (Ref. 09/2018). It was also necessary for the parents or legal guardians to provide informed consent, after having been informed of the possible risks of taking part, before the students could participate in the study.

*Physical education teachers*. Two physical education teachers with ample experience with contact sports and training in teaching methods were chosen to administer the two intervention programs. Each teacher imparted a different program. Both teachers knew the characteristics of the methodology that they were going to teach. Therefore, they paid special attention to presenting the tasks and the communication/feedback to the students according to the program to be imparted, to ensure its proper implementation. To this end, all sessions were recorded with a video camera (GoPro model), and the teachers’ voices were recorded with microphones (BY-LM10 model for Smartphones).

*Setting*. All the practical sessions of both intervention programs (TGAS and DIS) were given on an outdoor futsal pitch (40 m × 20 m), a common size for primary education schools in Spain. The pitch had a water fountain so that the students could hydrate themselves. The mean temperature during the sessions was 19 °C (min. 15 °C–max. 24 °C). Both intervention programs lasted three months (from April to June), with one or two weekly sessions. The school authorities indicated the days on which we could go to school to implement the intervention programs.

### 2.3. Variables

The study independent variables consisted of the TGAS program based on the TGA method, and the DIS program based on the DI method. Both intervention programs were designed in a homogenous manner for teaching football in school, but were based on different methodologies. A comparative analysis showed that both intervention programs were similar (*p* > 0.05) with a high degree of association in different variables, in terms of: (i) the number of tasks, (ii) the number of sessions, (iii) the phases of play, (iv) the specific contents, and (v) the didactic objectives; these were necessary requirements in order for the two programs to be considered similar. Due to the particularities of each teaching method, differences were identified in the rest of the studied variables, i.e., (vi) game situation, (vii) teaching means, (viii) level of opposition, (ix) degree of opposition, (x) density of the task, (xi) competitive load, and (xii) cognitive implication (*p* < 0.05) [49]. This made it possible to guarantee that the design of the programs was not a contaminating variable in the results obtained. Similarly, the tasks which made up each intervention program were validated by a panel of 13 experts, reaching excellent levels of validity (*Aiken’s V* ≥ 0.69) and internal consistency (*α* = 0.97) [50]. The results indicated that both intervention programs are valid and reliable for teaching football in school, as well as for comparing the effects of both methodologies.

A total of 30 dependent variables were recorded in the study, grouped into eTL and iTL variables. The eTL variables (objective way) used in the study were: (i) distance in meters (dis(m)); (ii) meters covered per minute (m/min); (iii) number of accelerations (Nacc); (iv) accelerations per minute (acc/min); (v) number of decelerations (Ndec); (vi) decelerations per minute (dec/min); (vii) maximum speed (MAX Speed); (viii) average speed (AVG Speed); (ix) percentage of time spent in high-intensity activities (HIA% ≥ 15 Km/h); (x) percentage of time spent walking (walk% ≤ 6km/h); (xi) percentage of time spent jogging (jog% = 6–12 km/h); (xii) percentage of time spent running (run% = 12–18 km/h); (xiii) percentage of time spent sprinting (sprint% = > 18 km/h); (xiv) number of sprints (Nsprints); (xv) number of impacts received (Nimpacts); (xvi) number of steps (Nsteps); (xvii) steps per minute (steps/min); (xviii) number of jumps (Njumps); (xix) jumps per minute (jumps/min); (xx) integral Player Load (PL); and (xxi) integral Player Load per minute (PL/min).

The variables of acc/min and dec/min were calculated following the formula proposed by Schelling et al. [51]:(1)$$\frac{\mathrm{acc}}{\min}=\frac{\sqrt{{\left(x_{n}-x_{n-1}\right)}^{2}+{\left(y_{n}-y_{n-1}\right)}^{2}+{\left(z_{n}-z_{n-1}\right)}^{2}}}{\min}$$

The PL variable was calculated from the accelerations using the formula proposed by Boyd et al. [29]:(2)$$\mathrm{PL}=\sqrt{\frac{{\left(a_{y1}-a_{y-1}\right)}^{2}+{\left(a_{x1}-a_{x-1}\right)}^{2}+{\left(a_{z1}-a_{z-1}\right)}^{2}}{10}}$$

The iTL variables (objective way) recorded in the study were: (i) maximum HR (HRmax); (ii) average HR (AVG HR); (iii) percentage of relative HR (rel HR%); (iv) percentage of time spent in the 50 to 60% HR range (50–60% HR); (v) percentage of time spent in the 60 to 70% HR range (60–70% HR); (vi) percentage of time spent in the 70 to 80% HR range (70–80% HR); (vii) percentage of time spent in the 80 to 90% HR range (80–90% HR); (viii) percentage of time spent in the 90 to 95% HR range (90–95% HR); and (ix) percentage of time spent in the 95 to 200% HR range (95–200% HR). The formula proposed by Tanaka et al. [52] was used to calculate maximum HR:(3)HRmax=208−0.7×age.

All of the analyzed variables allowed us to determine the variability of the activity performed. At a kinematic level, it was necessary to know the percentage of time that the students spent in different speed ranges, since they move at different speeds. These speeds determined whether a HIA was occurring. Regarding HR, it was necessary to know the percentage of time that the students spent in the different HR ranges. These HR ranges depended on the intensity of the activity performed.

The ratings of perceived exertion (RPE) (subjective iTL) shown by the students during the football practice in the assessment tests (pretest and posttest) were also analyzed.

### 2.4. Instruments

An inertial device called the WIMU Pro^TM^ (Real Track Systems, Almería, Spain) was used to measure the eTL and iTL (objectively) produced by the two intervention programs. Each device was synchronized with a GARMIN^TM^ HR monitor. The WIMU Pro^TM^ is an inertial recording device used to monitor physical activity and movement in real time. This inertial device uses different sensors to record the data (four accelerometers, a magnetometer, a gyroscope, GNSS, and UWB among others). The information recorded by the sensors is converted into quantitative data using the SPRO^TM^ software (Real Track Systems, Almería, Spain). The WIMU Pro^TM^ inertial device was used both in the school context [13] and in the field of sport training [35,53], and has proved to be valid and reliable [54,55].

The perceived effort (subjective iTL) recorded by the students themselves during football practice in the assessment tests was calculated using the Borg’s RPE scale [40], modified by Eston et al. [56]. These authors adapted the Borg’s RPE scale to the child population and inserted graphic illustrations (curvilinear pictorial scale) that represented the degree of perceived effort. Each illustration contains a level of effort that progressively increases from 0 to 10.

### 2.5. Procedure

First, it was necessary to obtain a series of authorizations, i.e., (i) the approval of the University Bioethics Committee (Ref. 09/2018); (ii) the authorizations of the school and the physical education teachers, (iii) the approval of the school council to include the intervention in the school study program, and (iv) the written informed consent of the students’ parents or legal guardians.

After obtaining the necessary approvals, an initial assessment (pretest) was made, in which the students played five 3 vs. 3 matches in a reduced space and with smaller goals. The figure of the goalkeeper was eliminated. The teams were mixed. Each match lasted five minutes and the students rested for two minutes between matches to indicate their RPE and freely hydrate themselves.

Then, the TGAS and DIS programs were applied, one to each class group, over 9 sessions. Every session was made up of four tasks with a duration of 10 minutes each. The sessions were of increasing complexity, beginning with simple tasks (1 vs. 0, 2 vs. 0, 1 vs. 1) and ending with more complex ones (3 vs. 2, 4 vs. 4, 5 vs. 5) [49]. The DIS program is characterized by the teaching of technical gestures isolated from the game that are integrated into real play situations once they are mechanized. The TGAS program is characterized by sport teaching through tasks that imply tactical knowledge, favoring decision-making on the part of the students. In all the sessions, the students had time to hydrate themselves between tasks.

Lastly, once the intervention programs had been applied to each class group, a final assessment (posttest) was performed where the students again played 3 vs. 3 matches with a game duration and rest intervals similar to those of the pretest. They also recorded their RPE. The organization of the students in the different teams was similar in both the pretest and posttest.

Figure 2 shows the distribution of the content for each methodological proposal.

At the beginning of each session, including the pretest and posttest assessments, every student was fitted with a WIMU Pro^TM^ inertial device placed in an anatomical harness and a HR monitor. Before placement, the inertial devices were calibrated and synchronized as proposed by Inglés-Bolumar et al. [35]. It took approximately 10 min to fit the devices, and together with the 45 min of physical activity, this accounted for the 55-min duration of the physical education classes.

### 2.6. Statistical Analyses

Firstly, the data recorded by the WIMU Pro^TM^ inertial devices were converted into kinematic variables using the SPRO^TM^ software. Then, a box and whisker plot was used to eliminate possible outliers. To be precise, only one case was eliminated because it recorded a speed greater than 35 km/h. After the elimination of the outliers, a cluster analysis was performed to establish the values for sprinting, HIA, and the speed ranges (walk, jog, run, and sprint), adapting them to the students’ characteristics. The SPRO^TM^ software was personalized with a specific data analysis configuration according to the ranges obtained from the cluster analysis. This procedure was necessary given that the software was configured by default to consider sprints as speeds of over 21 km/h. It is difficult for students to attain this velocity in physical education classes because of their physical characteristics and the play spaces used. After configuring the software, the data were again quantified for the statistical analysis. Sprints were considered speeds of 18 km/h or more.

Once the data had been quantified and converted into kinematic variables, Kolmogorov-Smirnov (normality), Rachas (randomness), and Levene (homogeneity) [57] tests were performed to determine the use of parametric or nonparametric statistical tests to confirm the hypothesis. The results obtained, as well as the decision to confirm the hypothesis, are presented in Table 1.

The *t*-test for independent samples was used for the parametric variables and the Mann-Whitney U test was used for the nonparametric variables [57]. The inferential analysis, according to gender, was only performed with the data recorded in the assessment tests (pre and posttest) because both class groups practiced the same tasks (3 vs. 3 matches). The analysis according to gender cannot be performed with the data recorded in the application of the programs because the loads were influenced by the tasks used in each program, i.e., TGAS and DIS.

A *t*-test for independent samples was used to analyze the RPE recorded in the assessment tests according to gender and teaching methodology. Then, a repeated measure ANOVA was performed to determine whether the TGAS and DIS programs influenced the RPE. Lastly, a *t*-test for related samples was used to study the evolution of each class group between the pretest and posttest, as well as the evolution of the boys and girls independently (by gender).

The effect size was calculated using Cohen’s d and partial eta squared [58].

## 3. Results

Figure 3 shows the descriptive results of the objective eTL and iTL variables studied in the application of the intervention programs and their percentage distribution. This figure, taking 50% as a reference, compares the means obtained for each variable according to the intervention program applied.

The descriptive results of the objective eTL and iTL variables studied in the assessment tests and their percentage distribution are presented in Figure 4. The data obtained in the pretest and posttest in the TGAS and DIS programs were grouped.

Table 2 presents the results of the differences between the TGAS and DIS programs applied.

The results of the differences in the assessments, according to gender and teaching methodology applied are shown in Table 3 and Table 4, respectively.

The results of the differences in subjective iTL (RPE) recorded in the assessment tests according to gender and teaching methodology are shown in Table 5. The repeated measures ANOVA showed that the TGAS and DIS intervention programs did not have any effect on RPE (*F* = 0.004; *p* = 0.949; *η2* = 0.000).

The *t*-test for related measures found no significant differences between the assessment tests in the boys (*t* = 1.411; *df* = 18; *p* = 0.175; *d* = 0.347). There were, however, significant differences between the assessment tests in the girls (*t* = 2.731; *df* = 15; *p* = 0.015*; *d* = 0.705). On the other hand, the *t*-test for related measures found significant differences between the assessment tests in the DIS program (*t* = 2.366; *df* = 17; *p* = 0.030*; *d* = 0.739). There were no significant differences between the assessment tests in the TGAS program (*t* = 1.785; *df* = 16; *p* = 0.093; *d* = 0.464). However, the students who participated in each intervention program presented a more efficient RPE when the programs had been completed (posttest).

## 4. Discussion

The objective of this study was to quantify and compare, according to gender and teaching methodology, the eTL, iTL, and RPE after the implementation of two intervention programs based on two different methodologies. The main findings indicate that the students who participated in the TGAS program recorded higher values of iTL, while those who participated in the DIS program recorded higher values of eTL and RPE (subjective iTL). Boys recorded higher eTL, iTL, and RPE values than girls in assessment tests. There was an evolution in the RPE of both class groups of students in the assessments, with a more efficient RPE being recorded at the end of the intervention (posttest).

*Objective eTL and iTL*. The design of the teaching tasks can influence the total distance covered, the number of high-intensity runs, and the different speed zones, and therefore, the physical and physiological demands. For Ballesta et al. [59], the variables that define the design and complexity of the tasks, like the play area, opposition, and type of game (teaching means), can define the eTL to which the students are exposed. Greater importance, therefore, should be given to the application of intervention programs (teaching sessions). The design of the tasks for each intervention program was different. The TGAS program presented tasks based on tactical learning, while the DIS program presented tasks based on technical learning.

Regarding the implementation of the intervention programs, the results indicate that the TGAS program involved more intensity compared to the DIS program, because the students who participated in it recorded more sprints and spent more time sprinting and performing high-intensity activities. This could mean that HRmax, AVG HR, and the percentage of time spent in the 95–200% HR range were also greater in the TGAS program.

On the other hand, the students who participated in the DIS program recorded more Nacc, acc/min, Ndec, and dec/min compared to those in the TGAS program, producing greater eTL (PL and PL/min). For Gaudino et al. [60], load is closely related to general activity and accelerations. The number of accelerations that were recorded in both programs could be due to the movement produced by executing the tasks, as well as the duration of these movements [31]. Similarly, Akenhead et al. [61] associate the decrease in the number of accelerations with an increase in the distance covered at slow speeds.

Following this line of thought, the TGAS program recorded lower MAX Speed and AVG Speed than the DIS program. The students participating in the DIS program (organized in rows and with tasks without opposition, and working on techniques) only covered a few meters, accelerating to the maximum from the beginning of the tasks and quickly reaching a high speed, without performing sprints in most of the actions due to the short distances covered, and then slowing down. These movements were performed repeatedly with short rests between them due to the organization in rows. This structure (in rows) could explain the fact that the students participating in the DIS program spent more time in the 80–90% HR range. Ballesta et al. [59] stated that exercises without opposition do not achieve a high cognitive load, but have an eTL which is accepted by students.

On the other hand, the students who participated in the TGAS program (with games that imply tactical knowledge and are with opposition) constantly moved around during the games, beginning with low intensity accelerations and progressively reaching high speeds [62]. The use of games produces more efficient and maximum intensity movements without needing to reach high speeds [63].

The students participating in the TGAS program spent more time in HIA due to the presence of defenders [59], thus reaching higher iTL values (HR). However, the students who participated in this program reached lower levels of eTL than those following the DIS program due to recording fewer accelerations. In the TGAS program, PL and PL/min did not show the same tendency as the rest of the variables studied because it is a concept that is closely related to the movements performed and changes in acceleration [64]. González-Espinosa et al. [13], in a study on teaching basketball in school, indicated that the tasks performed using the TGA method record higher values of iTL (HR), which is in agreement with the results obtained in this study. However, regarding eTL, the results do not agree, as those authors stated that eTL was also higher in the tasks performed following the TGA method. This discrepancy could be due to the type of tasks used, as well as to the sport studied, i.e., basketball. More research is warranted on the study of the physical and physiological demands resulting from the practice of contact sports in physical education classes.

Regarding the assessment tests (pretest and posttest) where the students from both groups carried out the same tasks (3 vs. 3 matches), the number of accelerations was not conditioned by the type of task used. In this case, the number of accelerations could be influenced by the level (or by the practice of out-of-school football) of the students. In this line of thought, Ingebrigtsen et al. [65] affirm that subjects with greater experience record fewer accelerations. In this study, 15% of the students who participated in the TGAS program practiced football in an out of school context, while 42.9% of those who participated in the DIS program practiced football out of school. Thus, the students who followed the DIS program recorded fewer accelerations. However, as in the DIS program, these students reached higher levels of eTL (PL and PL/min). Similarly, the students who participated in the TGAS program reached higher values of iTL (HR) in the assessment tests. In the context of sports training, Torres-Ronda et al. [66] pointed out that during SSGs, subjects with more experience show a greater eTL associated with a lower iTL, due to possible differences in fitness.

Gender differences indicated that boys recorded more Nacc, acc/min, Ndec, and dec/min than girls, leading to greater eTL (PL and PL/min). The boys also spent more time sprinting and performing high-intensity activities, while the girls spent more time walking. Thus, boys showed higher iTL (HR) values than girls in the assessment tests. Mckenzie et al. [67] indicated in their study that boys were more active than girls in general and during skill exercise and game play. Gender differences in physical activity could be due to students’ biological and motivational changes, social expectations (peer and teacher), and the development of skills. In order to promote physically-active classes, teachers must consider gender when planning physical education sessions [20].

*Subjective iTL (RPE).* Regarding the RPE recorded in the assessment tests, the students who participated in the TGAS program reported lower values of RPE in the pretest and posttest than those who followed the DIS program. Similarly, in spite of the fact that the results indicate that the intervention programs had no effects (*p* > 0.05) on the RPE of the students. There was an evolution in the RPE of the students of both class groups between the pretest and the posttest, with more efficient results being recorded in the posttest. Gender differences showed that boys gave higher values of RPE than girls in the pretest and posttest. There was an evolution in the RPE of both genders between the assessment tests. The evolution in the RPE of girls was significantly greater than that of boys (*p* < 0.05).

*Strengths and limitations*. Only one study [13] has investigated the eTL and iTL in primary education student after the application of intervention programs based on different teaching methodologies (TGA and DI). In this regard, our results also support the use of the methodologies that focus on an understanding of the game (as the TGA method).

The Spanish curriculum attaches great importance to the health of students through the teaching of physical education. The results of this study show that the TGA method implies a greater intensity compared to the DI method, favoring the physical fitness of the students, since it emphasizes aerobic fitness [18]. In this sense, aerobic fitness improvements are related to health enhancements [19]. Many students only practice physical activity at school. Therefore, physical education classes should be as intense as possible, favoring their health. For this reason, it is recommended that the TGA method be used.

This study presents several limitations: (i) the cognitive and physical characteristics of the students giving rise to specific differences; (ii) the mediating variables of students’ gender and experience (practice of out-of-school football); the academic authorities of the school organize the distribution of students in each class; and (iii) the duration of the physical education class which limits useful class time and makes it difficult to achieve the didactic objectives. These aspects cannot be controlled due to the peculiarities of the Spanish educational system, where academic groups (heterogeneous) are formed and the hours for physical education classes are assigned.

## 5. Conclusions

The results of this study show that eTL and iTL are determined by different parameters like play space, the presence of opposition, and the type of task used. Physical education teachers, based on these findings, can design teaching sessions and program loads according to didactic objectives. With regard to the study of the teaching methodologies, the results indicate that the TGA method enhances student physical fitness and health, so this method is recommended. Teachers must also consider gender, because boys register higher loads than girls.

## Figures and Tables

**Figure 1 ijerph-17-00344-f001:**
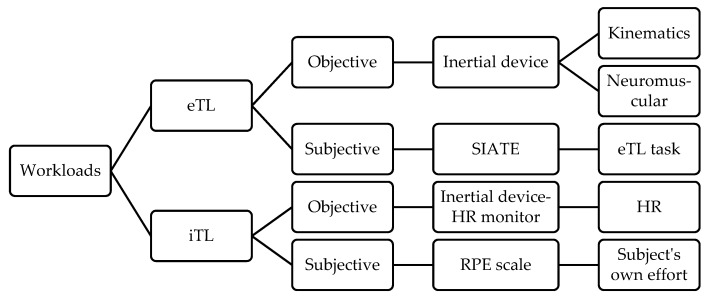
Classification of the workloads and measuring instruments. RPE, Ratings of Perceived Exertion; eTL, external load; iTL, internal load; HR, heart rate; SIATE (acronym in Spanish), Integral System for the Analysis of Training Tasks.

**Figure 2 ijerph-17-00344-f002:**
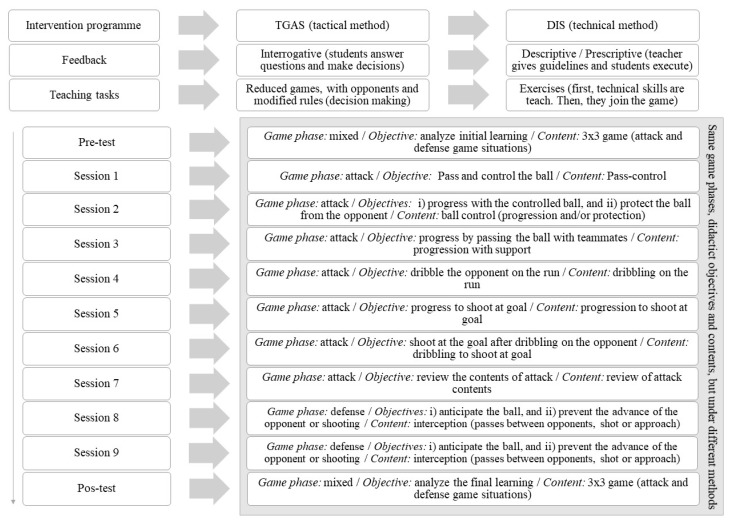
Distribution of the content for each methodological proposal.

**Figure 3 ijerph-17-00344-f003:**
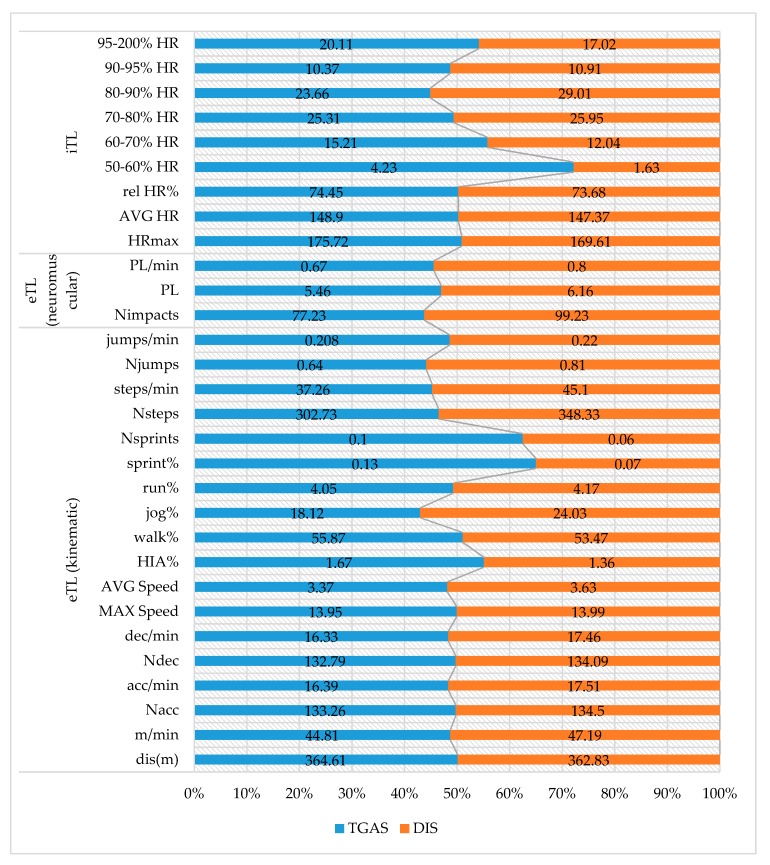
Descriptive analysis of the objective eTL and iTL variables in the application of the programs. Note: TGAS, Tactical Games Approach Soccer; DIS, Direct Instruction Soccer.

**Figure 4 ijerph-17-00344-f004:**
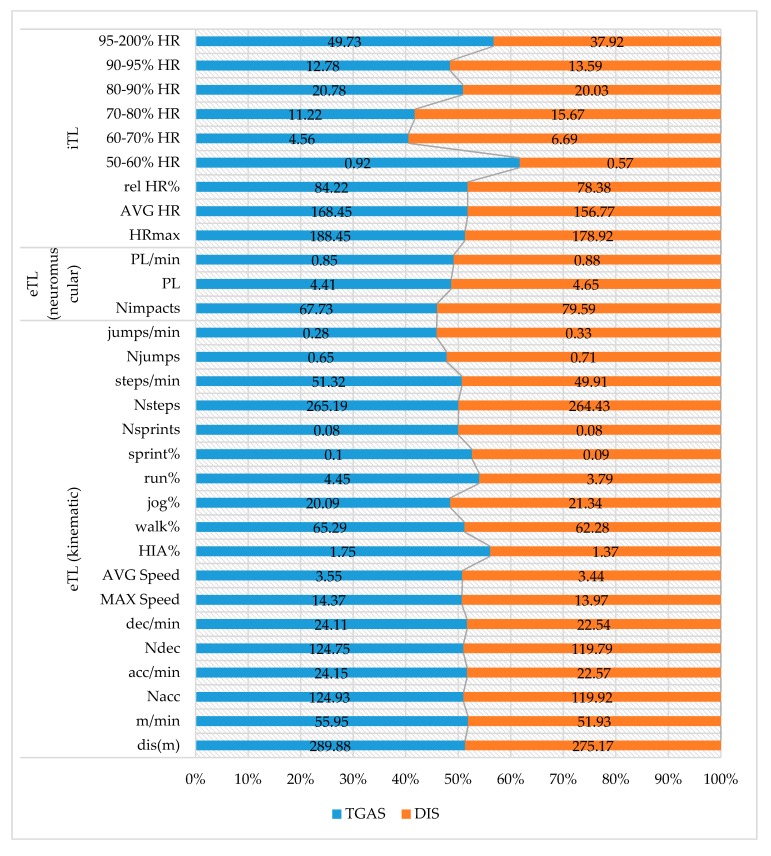
Descriptive analysis of the objective eTL and iTL variables in the assessment tests. Note: TGAS, Tactical Games Approach Soccer; DIS, Direct Instruction Soccer.

**Table 1 ijerph-17-00344-t001:** Results of the statistical tests for the assumption of the following criteria.

Study Variables		Application of the Programs (Sessions)							Pre and Post (Assessment Tests)						
		*K-S*		*Rachas*		*Levene*		Test	*K-S*		*Rachas*		*Levene*		Test
Kinematic eTL	dis(m)	0.000	*	0.000	*	0.000	*	N-P	0.200 ^a^		0.001	*	0.186		N-P
	m/min	0.036	*	0.000	*	0.276		N-P	0.021	*	0.001	*	0.050		N-P
	Nacc	0.008	*	0.000	*	0.709		N-P	0.200 ^a^		0.123		0.008	*	N-P
	acc/min	0.036	*	0.000	*	0.000	*	N-P	0.200 ^a^		0.004	*	0.149		N-P
	Ndec	0.009	*	0.000	*	0.689		N-P	0.200 ^a^		0.174		0.009	*	N-P
	dec/min	0.048	*	0.000	*	0.000	*	N-P	0.200 ^a^		0.014	*	0.151		NP
	MAX Speed	0.002	*	0.000	*	0.000	*	N-P	0.200 ^a^		0.324		0.927		P
	AVG Speed	0.058		0.000	*	0.389		N-P	0.019	*	0.031	*	0.717		N-P
	HIA%	0.000	*	-		0.147		N-P	0.000	*	-		0.050		N-P
	walk%	0.009	*	0.000	*	0.002	*	N-P	0.200 ^a^		0.324		0.720		P
	jog%	0.186		0.000	*	0.435		N-P	0.096		0.006	*	0.278		N-P
	run%	0.000	*	0.000	*	0.312		N-P	0.000	*	0.114		0.038	*	N-P
	sprint%	0.000	*	-		0.001	*	N-P	0.000	*	-		0.585		N-P
	Nsprints	0.000	*	-		0.000	*	N-P	0.000	*	-		0.651		N-P
	Nsteps	0.000	*	0.000	*	0.052		N-P	0.004	*	0.000	*	0.013	*	N-P
	steps/min	0.000	*	0.000	*	0.994		N-P	0.001	*	0.000	*	0.006	*	N-P
	Njumps	0.000	*	-		0.030	*	N-P	0.000	*	-		0.047	*	N-P
	jumps/min	0.000	*	0.003	*	0.189		N-P	0.000	*	0.000	*	0.026	*	N-P
Neuromus-cular eTL	Nimpacts	0.000	*	0.000	*	0.145		N-P	0.009	*	0.000	*	0.506		N-P
	PL	0.000	*	0.000	*	0.057		N-P	0.200 ^a^		0.000	*	0.412		N-P
	PL/min	0.000	*	0.000	*	0.756		N-P	0.200 ^a^		0.000	*	0.366		N-P
iTL	HRmax	0.000	*	0.040	*	0.690		N-P	0.000	*	0.019	*	0.023	*	N-P
	AVG HR	0.000	*	0.143		0.373		N-P	0.000	*	0.370		0.085		N-P
	rel HR%	0.000	*	0.143		0.373		N-P	0.000	*	0.370		0.085		N-P
	50–60% HR	0.000	*	-		0.000	*	N-P	0.000	*	-		0.124		N-P
	60–70% HR	0.000	*	0.112		0.138		N-P	0.000	*	-		0.084		N-P
	70–80% HR	0.000	*	0.007	*	0.001	*	N-P	0.000	*	0.369		0.392		N-P
	80–90% HR	0.000	*	0.640		0.000	*	N-P	0.000	*	0.618		0.002	*	N-P
	90–95% HR	0.000	*	0.003	*	0.000	*	N-P	0.000	*	0.549		0.583		N-P
	95–200% HR	0.000	*	0.004	*	0.191		N-P	0.000	*	0.764		0.037	*	N-P

Note: K-S, the Kolmogorov-Smirnov test to measure normality; Rachas, the Rachas test for randomness; Levene, the Levene test for homogeneity of variance; P, parametric test; N-P, nonparametric test; ^a^ 0.200 in K-S test, Lower limit of true significance; * *p* < 0.05.

**Table 2 ijerph-17-00344-t002:** Results of the application of the programs according to the teaching methodology.

Study Variables		TGAS/ *M* ± *SD*	DIS/ *M* ± *SD*	*U*	*p*		*d_Cohen_*
Kinematic eTL	dis(m)	364.61 ± 143.44	362.83 ± 113.16	145,025.000	0.630		0.029
	m/min	44.81 ± 13.82	47.19 ± 13.55	133,634.500	0.007	*	0.163
	Nacc	133.26 ± 57.68	134.50 ± 55.93	146,650.000	0.867		0.010
	acc/min	16.39 ± 6.02	17.51 ± 7.04	137,734.500	0.059		0.115
	Ndec	132.79 ± 57.82	134.09 ± 56.14	146,566.500	0.854		0.011
	dec/min	16.33 ± 6.05	17.46 ± 7.07	137,679.500	0.057		0.115
	MAX Speed	13.95 ± 3.33	13.99 ± 2.58	142,996.500	0.382		0.053
	AVG Speed	3.37 ± 0.73	3.63 ± 0.73	120,076.000	0.000	*	0.326
	HIA%	1.67 ± 2.82	1.36 ± 2.98	134,827.000	0.005	*	0.149
	walk%	55.87 ± 13.28	53.47 ± 14.35	134,603.000	0.013	*	0.152
	jog%	18.12 ± 10.60	24.03 ± 10.70	102,498.000	0.000	*	0.547
	run%	4.05 ± 4.45	4.17 ± 5.05	146,777.500	0.885		0.009
	sprint%	0.13 ± 0.52	0.07 ± 0.51	140,593.500	0.004	*	0.081
	Nsprints	0.10 ± 0.30	0.06 ± 0.23	140,788.000	0.005	*	0.079
	Nsteps	302.73 ± 176.90	348.33 ± 158.03	118,470.500	0.000	*	0.345
	steps/min	37.26 ± 19.72	45.10 ± 19.70	111,538.500	0.000	*	0.431
	Njumps	0.64 ± 1.00	0.81 ± 1.18	135,990.000	0.013	*	0.135
	jumps/min	0.21 ± 0.14	0.23 ± 0.15	24,354.500	0.094		0.155
Neuromus-cular eTL	Nimpacts	77.23 ± 62.07	99.23 ± 69.79	114,177.500	0.000	*	0.398
	PL	5.46 ± 2.56	6.16 ± 2.32	119,110.500	0.000	*	0.338
	PL/min	0.67 ± 0.28	0.80 ± 0.28	108,586.500	0.000	*	0.469
iTL	HRmax	175.72 ± 21.16	169.61 ± 24.07	115,053.500	0.000	*	0.233
	AVG HR	148.90 ± 20.78	147.37 ± 22.04	131,202.500	0.730		0.021
	rel HR%	74.45 ± 10.39	73.68 ± 11.02	131,202.500	0.730		0.021
	50–60% HR	4.23 ± 12.09	1.63 ± 6.18	120,060.500	0.000	*	0.167
	60–70% HR	15.21 ± 19.55	12.04 ± 19.48	115,351.500	0.000	*	0.229
	70–80% HR	25.31 ± 19.73	25.95 ± 23.07	130,436.000	0.613		0.031
	80–90% HR	23.66 ± 17.16	29.01 ± 22.97	118,374.500	0.002	*	0.189
	90–95% HR	10.37 ± 10.88	10.91 ± 13.57	128,038.500	0.309		0.063
	95–200% HR	20.11 ± 26.54	17.02 ± 26.29	118,988.000	0.003	*	0.181

Note: *M*, Mean; *SD*, Standard Deviation; TGAS, Tactical Games Approach Soccer; DIS, Direct Instruction Soccer; dis(m), meters in the entire program; * *p* < 0.05.

**Table 3 ijerph-17-00344-t003:** Results of the assessment tests according to gender.

Study Variables		Boys/ *M* ± *SD*	Girls/ *M* ± *SD*	*U/t*	*p*		*d_Cohen_*
Kinematic eTL	dis(m)	289.31 ± 64.04	275.37 ± 70.80	18,225.500 ^a^	0.017	*	0.236
	m/min	55.11 ± 12.35	52.78 ± 13.24	18,681.000 ^a^	0.046	*	0.198
	Nacc	126.93 ± 28.64	117.42 ± 26.40	17,199.000 ^a^	0.001	*	0.322
	acc/min	24.13 ± 5.17	22.54 ± 5.05	17,629.500 ^a^	0.004	*	0.286
	Ndec	126.75 ± 28.67	117.28 ± 26.40	17,220.000 ^a^	0.001	*	0.321
	dec/min	24.09 ± 5.18	22.52 ± 5.05	17,648.500 ^a^	0.004	*	0.284
	MAX Speed	14.67 ± 2.65	13.61 ± 2.62	4.081 ^b^	0.000	*	0.403
	AVG Speed	3.59 ± 0.56	3.38 ± 0.55	16,606.500 ^a^	0.000	*	0.373
	HIA%	1.73 ± 2.74	1.34 ± 2.56	18,835.000 ^a^	0.034	*	0.185
	walk%	62.71 ± 9.31	65.20 ± 8.94	-2.757 ^b^	0.006	*	0.272
	jog%	22.23 ± 8.64	18.88 ± 7.96	16,284.000 ^a^	0.000	*	0.401
	run%	4.62 ± 4.02	3.56 ± 4.09	17,045.500 ^a^	0.001	*	0.335
	sprint%	0.16 ± 0.60	0.02 ± 0.13	19,249.000 ^a^	0.001	*	0.151
	Nsprints	0.12 ± 0.34	0.03 ± 0.17	19,283.000 ^a^	0.001	*	0.148
	Nsteps	287.74 ± 97.03	238.06 ± 93.26	14,885.000 ^a^	0.000	*	0.525
	steps/min	54.87 ± 19.01	45.73 ± 18.32	15,159.000 ^a^	0.000	*	0.500
	Njumps	0.72 ± 0.99	0.63 ± 0.92	20,067.000 ^a^	0.342		0.084
	jumps/min	0.31 ± 0.17	0.30 ± 0.16	3717.000 ^a^	0.580		0.080
Neuromus-cular eTL	Nimpacts	87.68 ± 50.75	56.66 ± 33.61	13,196.000 ^a^	0.000	*	0.682
	PL	5.06 ± 1.45	3.91 ± 1.20	10,942.500 ^a^	0.000	*	0.912
	PL/min	0.96 ± 0.29	0.75 ± 0.24	11,360.500 ^a^	0.000	*	0.867
iTL	HRmax	185.42 ± 28.24	182.25 ± 23.70	17,509.000 ^a^	0.024	*	0.227
	AVG HR	163.54 ± 26.47	162.18 ± 22.78	19,024.000 ^a^	0.337		0.096
	rel HR%	81.77 ± 13.24	81.09 ± 11.39	19,024.000 ^a^	0.337		0.096
	50–60% HR	1.10 ± 6.10	0.36 ± 2.33	19,451.500 ^a^	0.133		0.059
	60–70% HR	6.02 ± 15.31	5.08 ± 15.34	18,382.500 ^a^	0.50		0.151
	70–80% HR	13.77 ± 16.92	12.85 ± 18.51	18,494.500 ^a^	0.143		0.141
	80–90% HR	17.93 ± 16.92	23.21 ± 18.13	16,433.500 ^a^	0.001	*	0.322
	90–95% HR	12.20 ± 11.43	14.24 ± 12.34	18,452.500 ^a^	0.146		0.145
	95–200% HR	46.14 ± 34.42	41.86 ± 32.85	18,541.000 ^a^	0.168		0.137

Note: *M*, Mean; *SD*, Standard Deviation; ^a^ Mann-Whitney U-test; ^b^
*t*-test for independent samples; dis(m), meters in the entire program; * *p* < 0.05.

**Table 4 ijerph-17-00344-t004:** Results of the assessment tests according to teaching methodology.

Study Variables		TGAS/ *M* ± *SD*	DIS/ *M* ± *SD*	*U/t*	*p*		*d_Cohen_*
Kinematic eTL	dis(m)	289.88 ± 69.03	275.17 ± 65.12	19,081.500 ^a^	0.084		0.171
	m/min	55.95 ± 13.20	51.93 ± 12.03	17,845.500 ^a^	0.006	*	0.274
	Nacc	124.93 ± 23.63	119.92 ± 32.00	18,948.000 ^a^	0.066		0.182
	acc/min	24.15 ± 4.69	22.57 ± 5.55	17,921.500 ^a^	0.007	*	0.267
	Ndec	124.75 ± 23.64	119.79 ± 32.02	18,947.500 ^a^	0.066		0.182
	dec/min	24.11 ± 4.69	22.54 ± 5.55	17,960.000 ^a^	0.008	*	0.264
	MAX Speed	14.37 ± 2.62	13.97 ± 2.74	1.517 ^b^	0.130		0.150
	AVG Speed	3.55 ± 0.54	3.44 ± 0.58	19,367.500 ^a^	0.136		0.147
	HIA%	1.75 ± 2.85	1.37 ± 2.43	19,981.000 ^a^	0.265		0.097
	walk%	65.29 ± 9.03	62.29 ± 9.18	3.341 ^b^	0.001	*	0.330
	jog%	20.09 ± 8.14	21.34 ± 8.83	18,809.500 ^a^	0.051		0.193
	run%	4.45 ± 4.37	3.79 ± 3.72	19,674.500 ^a^	0.214		0.122
	sprint%	0.10 ± 0.40	0.09 ± 0.50	20,901.000 ^a^	0.634		0.022
	Nsprints	0.08 ± 0.27	0.08 ± 0.28	20,925.000 ^a^	0.664		0.020
	Nsteps	265.19 ± 105.97	264.43 ± 89.51	20,257.500 ^a^	0.451		0.074
	steps/min	51.32 ± 21.00	49.91 ± 17.09	21,022.500 ^a^	0.904		0.012
	Njumps	0.65 ± 0.87	0.71 ± 1.05	21,132.000 ^a^	0.973		0.003
	jumps/min	0.28 ± 0.14	0.33 ± 0.19	3324.000 ^a^	0.065		0.270
Neuromus-cular eTL	Nimpacts	67.73 ± 44.49	79.59 ± 47.56	17,509.500 ^a^	0.002	*	0.302
	PL	4.41 ± 1.48	4.65 ± 1.42	17,616.500 ^a^	0.003	*	0.293
	PL/min	0.85 ± 0.29	0.88 ± 0.28	18,593.000 ^a^	0.033	*	0.211
iTL	HRmax	188.45 ± 20.22	178.92 ± 30.82	16,826.500 ^a^	0.004	*	0.288
	AVG HR	168.45 ± 20.31	156.77 ± 27.70	14,814.500 ^a^	0.000	*	0.470
	rel HR%	84.22 ± 10.16	78.38 ± 13.85	14,814.500 ^a^	0.000	*	0.470
	50–60% HR	0.92 ± 5.84	0.57 ± 3.04	19,667.000 ^a^	0.292		0.041
	60–70% HR	4.56 ± 13.46	6.69 ± 17.09	17,316.000 ^a^	0.002	*	0.245
	70–80% HR	11.22 ± 18.41	15.67 ± 16.56	14,783.500 ^a^	0.000	*	0.473
	80–90% HR	20.78 ± 19.34	20.03 ± 15.68	19,830.000 ^a^	0.782		0.027
	90–95% HR	12.78 ± 11.57	13.59 ± 12.27	19,546.500 ^a^	0.603		0.052
	95–200% HR	49.73 ± 34.97	37.92 ± 31.20	16,226.000 ^a^	0.001	*	0.341

Note: *M*, Mean; *SD*, Standard Deviation; TGAS, Tactical Games Approach Soccer; DIS, Direct Instruction Soccer; ^a^ Mann-Whitney U-test; ^b^
*t*-test for independent samples; dis(m), meters in the entire program; * *p* < 0.05.

**Table 5 ijerph-17-00344-t005:** Results of the RPE scale in the assessment tests according to gender and teaching methodology.

**Gender**	**Boys/*M* ± *SD***	**Girls/*M* ± *SD***	***t***	***df***	***p***		***d_Cohen_***
Pretest	3.12 ± 1.15	3.09 ± 1.43	0.071	36	0.944		0.023
Posttest	2.81 ± 0.76	2.30 ± 0.77	2.041	36	0.049	*	0.666
**Methodology**	**TGAS/*M* ± *SD***	**DIS/*M* ± *SD***	***t***	***df***	***p***		***d_Cohen_***
Pretest	2.66 ± 1.39	3.56 ± 0.96	2.308	36	0.027	*	0.749
Posttest	2.21 ± 0.91	2.92 ± 0.50	3.034	36	0.004	*	0.986

Note: *M*, Mean; *SD*, Standard Deviation; *df*, degrees of freedom; TGAS, Tactical Games Approach Soccer; DIS, Direct Instruction Soccer; * *p* < 0.05.

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
