# Peer review of "Quantification of Internal and External Load in School Football According to Gender and Teaching Methodology"

_ijerph, 2020, doi:10.3390/ijerph17010344_

Round 1

Reviewer 1 Report

The purpose of this study was to quantify the external (eTL) and internal (iTL) load produced by the application of two different teaching methodologies: Tactical Games Approach (TGA) and Direct Instruction (DI). The paper is very interesting, nevertheless, I am going to propose some suggestions and comments in order to do easier the reading and quality of it.

Specific comments:

Methods

Please clarify the design of the study, it is confuse.

Why the participants are not distributed in the same number of players’ gender.

What about the out of school activity during the application of the programs? Could the results be affected by this?

In a part of the paper, the authors affirm, “Both intervention programs lasted three months, with two weekly sessions”, but after appears in the text “Then, the TGAS and DIS programs were applied, one to each class group, over 9 sessions”, is not it a contraction?

It is not clear how the distribution of the contents for each methodology proposals is? I think it needs more explanation.

Was there any outlier in other variables?

It is not clear, “However, the students who participated in both intervention programs presented a more efficient RPE when the programs had been completed (post-test).”, there were students who participated in both programs?

Why did not the authors use a statistical approach in order to combine gender and methodology variables?

Results

The data from the figures 3 and 4 are not clear, percentage about what? Please explain a little more or use other type of charts, means and standard deviations are recommended.

The values from the tables 2 and 3 are confuse. For instance what means 364.61 m for the variable dis(m)? in a session? In the entire program? In a task?

This idea is not clear, please rewrite: “The study by González-Espinosa et al. [13] together with this one are the only investigations to have studied eTL and iTL in students of primary education after the application of intervention programs based on different teaching methodologies, TGA and DI, justifying the use of the methodologies that focus on an understanding of the game.

The last conclusion I think also has to be reviewed “With regard to the study of the teaching methodologies, the results indicate that the TGA method favors the students’ physical fitness and integral development,” because the integral development has not been studied in this study.

Author Response

All manuscript
- First, a native translator performed a grammatical revision of the manuscript.
- All corrections were marked in yellow.
- Different paragraphs were rewritten to facilitate the reading of the manuscript. In addition, different verbs were changed by other more suitable verbs.
--------------------------------
Introduction
- The concept of ‘invasion sport’ was defined (lines 35-37).
- It was explained how subjective eTL is calculated through the SIATE instrument (lines 90-93).
--------------------------------
Methods
- The study design was clarified (lines 124-127). For this purpose, Figure 2 was also introduced (line 241-243).
- It was explained why the participants were not distributed in the same number of students by gender (This distribution depends on the authorities of the Spanish education system) (lines 131-133).
- It was explained that neither student participated in both intervention programs (line 137). It was also explained that the sample was not modified to maintain the ecological validity of the study (line 142).
- It was explained that the two Physical Education teachers selected to teach each intervention program knew the characteristics of the methodology on which it was based. Therefore, both teachers were selected (153-154).
- The months of study were clarified (lines 162-164).
- It was indicated which authors designed the RPE pictorial scale used in the study (lines 218-219).
- It was clarified that the teams that participated in the 3vs3 matches (assessment tests) were mixed (lines 227).
- It was clarified how the data was collected (lines 244-247). The ‘instruments section’ explains their operation and applicability.
- It explains why the outlier indicated in the study was eliminated (lines 253-254).
- Study hypotheses were defined (lines 266-268).
--------------------------------
Results
- The data in Figures 4 and 5 was clarified. The ‘means’, according to the method, of the study variables were compared (lines 285-286).
- In the Tables 2, 3 and 4, it was clarified that the ‘variable dis(m)’ refers to the entire intervention program.
--------------------------------
Discussion
- It was clarified that the tasks used in each intervention program were different (TGAS=tactical tasks; DIS=technical tasks) (lines 330-332).
- It was explained why teachers should attend to gender when planning physical education classes (lines 386-389).
- It was explained why teachers should use the TGA method in their physical education classes (lines 403-407).

Reviewer 2 Report

Dear Authors,

The article Quantification of internal and external load in school football according to gender and teaching methodology is a quite interesting piece of work, espcialy for the practitioners, footbal trainers mostly, probably for the teachers, and physical education methodists.

That's why my first thought was - the Authors missed the target journal...

Authors applied difficult, effort demanding methods (well described methods and research procedure), but still the sample size is questionable, even in pre-experimental study, as well as it looks that finally the limitations outweight the conclusions.

I recommend to reject this paper, and suggest to the Autors to change the target journal for, i.e.:

Physical Education and Sport Pedagogy  European Physical Education Review Journal of Teaching in Physical Education Advances in Physical Education

Kind regards

Author Response

(The authors gave the same response as above.)

Reviewer 3 Report

However, the research gap based on existing body of literature is not clearly presented. What it was the  novelty of study?

  Since the discussion is not opened to the reader, it is difficult to see how your study moves this field forward and what is the contribution you aim to make to the theory. This can also be seen in the discussion section where the theoretical and practical implications are omitted.

 You need to know the discussion and position your paper accordingly against this existing body of knowledge.

 The theoretical section should be able to explain the topic as well,  to understand the phenomenon under study.

Material and methods: I did not understand too well how the participants were selected?  What was the inclusion and exclusion criterias of your study?

  Please describe what “you did” in practice, the method for data collection nor how they have been used in the analysis.

 The theoretical and practical contributions derived from the results are not presented in the paper.

The discussion section does not help to identify what the results provided in the light of existing body of knowledge that could bring the discussion forward. In Discussions, you need to highlight the correlation of your results with previous studies. You have only 10 references for this section.

 Authors don’t give an answer on crucial question “why do in this context gender differences matter”.

 How could the findings be theoretically generalized and therefore also applied to other contexts?

I hope the comment will help you to improve tour paper. 

Author Response

(The authors gave the same response as above.)

Reviewer 4 Report

Dear Author(s)

This manuscript was titled as "Quantification of internal and external load in school football according to gender and teaching methodology", but Intro failed to setup the environment for the reader. Moreover, the research design and the methods used were not appropriate to answer your RQ. A series of T-tests and other stats were used, while this design is a 2X2 mixed factorial ANOVA. Moreover, maybe a crossover design would have been more appropriate. 

Author(s) claimed that results support their hypothesis, but nowhere in the manuscript a hypothesis was stated. RQ was attempted to be tested using 2 groups that had different characteristics, different proportion of gender and different applied workloads employed by the two different pedagogical teaching approaches. 

In addition, many physiological elements were misrepresented and misused like RPE and HR. In other words this study claimed that TAG and DIS was responsible for increasing HR differently in boys and girls by running simple t-tests failing to run  a 2X2. Moreover, the workload applied in the TAG and DIS was different, even though in some places was states that it was the same.

Writing style needs significant improvement and proofreading from an English language speaker. I have noticed many issues that words were probably translated to English, failing to capture the correct meaning in English (e.g. invasion sports like football and basketball while the proper expression is contact sports). 

Please see my detailed review for more.

All the best

Author Response

(The authors gave the same response as above.)

Round 2

Reviewer 2 Report

Dear Authors,

In lines 129-142 - the Authors probably meant grade instead of year.

In lines 35-37 - if the Authors write "scientific literature...", to defence the using of term 'invasion sport', the Authors need to provide minimum 3 references, and especially English, international journal references, in case of the lack of such - the use of this term is unreasonable, and I'll stay with commonly use in international literature 'contact sports' term which has the same meaning following Authors’ explanation. (i.e. Farias C, Valério C, Mesquita I. Sport Education as a Curriculum Approach to Student Learning of Invasion Games: Effects on Game Performance and Game Involvement. J Sports Sci Med. 2018;17(1):56–65.) 

In the line 219 the Authors provide wrong reference - the source of the scale was a book by G. Borg (1998): Borg's perceived exertion and pain scales.; the cited source was the validation, critique and modification of the RPE Scale. Therefore, this is huge mistake or neglect!

Figure 2 is redundant and caused more mess to this part of the manuscript, I’m not follow why it was included?

Once again, the Figure 3 is not necessary to include it as a figure, but just describe it in the text.

The main issue for this manuscript is, that after the review process, and so many comment been sent, still in lines 418-421 Authors conclude: "With regard to the study of the teaching methodologies, the results indicate that the TGA method favors the students’ physical fitness and health, so this method is recommended when planning Physical Education sessions" - the Authors did not asses children's fitness and health at all, as well as that assessment was not the aim in this study. 

This conclusion needs to be redefined and match to the aims and hypothesis which is a great methodological issue.

The manuscript contains many minor shortcomings that the Authors had a chance to correct after the first reviews.

And once again, I suggest to the Authors the change of the target journal.

The text

Kind regards

Author Response

All corrections were marked in yellow.
- In lines 140-154, we described the study sample: i) total participants (elementary students of a school in Spain), distributed in two class group (fith year of group A, and fith year of group B). Each class group participated in a different intervention program (5A  Tactical Games Approach Soccer and 5B  Direct Instruction Soccer); ii) the age range of the participants, the average age (M) and the standard deviations (SD); and iii) the percentage of students in each class group who practiced football as an out of school activity during the application of the programs. In addition, it was indicated that the distribution of the students by gender in each class in the Spanish state system is mixed and heterogeneous and it depends on the school’s academic authorities. The class groups were not modified to maintain the ecological validity of the study.
- The term ‘invasion sports’ was replaced by ‘contact sports’. This correction was indicated by several reviewers.
- In lines 235-238, it was indicated that the RPE pictorical scale proposed by Eston and Parfitt (2007) is an adaptation for child population of the RPE scale designed and validated by Borg (1998).
- Figure 2 was included because other reviewers requested information on the distribution of content for each methodological proposal. This figure shows the contents worked on the sessions that constitute each intervention program, DIS and TGAS.
- Figure 3 was eliminated, although it was indicated that a box and whisker plot was used to eliminate posible outliers. In addition, it was specified that only one case was eliminated for exceeding 35 km/h. Later, it was indicated why the students could not reach these speeds (lines 271-280).

Reviewer 3 Report

Th authors improved the manuscript according with our recommendations.

Author Response

Thanks for your review.

Reviewer 4 Report

Dear Authors

Thanks for the revisions and the improved in English version, but still this manuscript is not in a condition to be published. See my detailed review, but briefly let me explain my point of view.

First, a similar article with pretty much the same design on a different sport was recently published by the same group at Journal of Strength and Conditioning Research, 11 Sep 2018. 

This of course doesn't justify this current attempt as one that has the merit to be published, as I detected series of flaws in the design, methodology, terminology etc. 

Starting from the abstract, you claim that "The use of the TGA method favors the students’ physical fitness and health, and thus, this method is recommended when planning Physical Education sessions." But how can you make such claims, when none of the examined variables is part fo the fitness components nor any of the tested variables is related to health. Your design was not setup to test as such.  

Moreover, what is internal and external load - the use of this terminology -  is not explained or cited in this manuscript adequately nor in the Journal of Strength and Conditioning Research, 11 Sep 2018. Who coined that term and what was the context is important to be stated. Using such terms without a proper established prior definition is improper. 

It is seems to be a confusion in regards to physical - physiological demands and the use of these terms. It is different to state physical and physiological compared to physical-physiological demands. I provided some definitions in my review.

Similar problem I experienced with the RPE and used scale. Authors claimed that they used the RPE scale but they cite Drs. Eston and Parffitt. Drs Eston and Parffitt actually did modify the Borg RPE scale to one that is more suitable for kids that has different name and no credit is given. This info is incorrect and doesn't represent their work. This is a serious issue in academia.

Another issue is with invasion sports and the use of the term games and sports interchangeably like to be the same. There is no invasion sports, there are contact sports and games exercises that we use to teach and practice the sports.

Moving to sections, Intro fails to deliver the message especially in regards to the gender relevance. Hypothesis is not stated in the right place, at the end of the Intro for example and it is too confusing. Authors stated "H1: The TGA method causes higher levels of eTL, iTL and RPE than the DI method. H2: Boys recorded higher 267 levels of eTL, iTL and RPE than girls."

But eTL has 21 parameters and iTL has 9 parameters which some of them were significant and some were not. Where do the authors draw the line between all of these variables of rejecting the null hypothesis? It is impossible to come to a conclusion having all of these variables.

Information that belongs to different sections is misplaced to another.

Methods is better than the last time, but now more serious methodological flaws were emerged. This is located in the different applied loads based on the teaching approaches and effect of gender and previous experience that was not controlled. Stats are confusing and improper use of statistical methods to answer the RQ were employed. In addition, data from an instrument were clustered to fit a population that this instrument was not validated for such use. 

Results are chaotic, reader is lost in a gazillion of reported data that do not add anything extra to the purpose of the study and do not prove the point. Based on the 21 variables of eTL and 9 iTL (that some were sig and some not) authors made claims that "The use of the TGA method favors the students’ physical fitness and health, and thus, this method is recommended when planning Physical Education sessions." But Authors did not examine fitness components not health parameters so such claims are unsubstantiated.

I do not thing that any other revision will be able to fix these issues and provide results and significance to the study that worths the merit of publication.

All the best

Author Response

All corrections were marked in yellow.
- A difference from the study published by the same group at Journal of Strength and Conditioning Research, this study took into account gender and experience (practice football outside of school during the application of the programs) when studiying the physical condition of primary education students. Therefore, the novelty of this study was that the designs should take into account gender and experience when working in mixed and heterogeneous contexts.
- The term ‘invasion sports’ was replaced by ‘contact sports’. This correction was indicated by several reviewers.
- It was justified that the use of SCAs (e.g. the TGA method) favors the physical fitness of students, since the improve aerobic resitance (Mancha et al., 2017). Also, an increase in aerobic resistance favors health (Castillo and Del Corral, 2010) (lines 66-67).
- More information about the terms ‘external load (eTL)’ and ‘internal load (iTL)’ is provided, in order to facilitate understanding of these terms (lines 80-86).
- The study hypotheses were inserted at the end of the ‘introduction section’ (lines 129-131).
- In the ‘sample section’, it was indicated that the Spanish education system prohibits segregating students by gender (line 144).
- In the ‘variables section’, it was explained why the described variables were used, which determine the variability of the activity performed (lines 217-221).
- In the ‘instrument section’, different studies were indicated that used the WIMU ProTM inertial devices both in the school context (González-Espinosa et al. 2013) and in the field of sport training (Inglés-Bolumar et al., 2018; Reina et al., 2019). Studies that describe the validity and reliability of these inertial devices were also indicated (Bastida-Castillo et al., 2018; Hernández-Belmonte et al., 2019) (lines 231-233).
- In lines 235-238, it was indicated that the pictorical scale proposed by Eston and Parfitt (2007) is an adaptation for child population of the RPE scale designed and validated by Borg (1998).
- Figure 3 was eliminated, although it was indicated that a box and whisker plot was used to eliminate posible outliers. In addition, it was specified that only one
case was eliminated for exceeding 35 km/h. Later, it was indicated why the students could not reach these speeds (lines 271-280).

Round 3

Reviewer 4 Report

Dear author(s), in general, I feel that this study has serious methodological flaws like design and statistics used to answer the research questions. This was not an air-tight design. It had plenty limitations influencing the outcomes that could have been controlled but they were not. I found plenty of occasions were claims were made that couldn’t be supported either by the findings or the references provided. I read plenty of the provided references and I couldn’t find any evidence to support authors’ claims. This is serious in academic writing as false information is disseminated. Operational definitions are missing, tables and figures have incomplete information, as none of these can stand-alone without any reference to the text. I noticed many cases where there one sentence to be one paragraph and improper use of English language to convey the message (e.g. research design to contaminate the outcome). Intro some how sets the topic but still is missing to lead the reader to understand the significance of this study. Methods do have a proper flow and many issues with the design and the abundance of the examined variables create a confusion. Stats are inappropriate. Results are confusing and difficult to follow. Discussion is impossible to supported as not all of the examined variables are discussed. Authors decided to measure and report 30 variables but they only discuss few of them. Moreover, they tried to group them and generalize them, but Results section notes different for some variables. For more, please see my 125 comments.